# Kinetics of Biodiesel Production from Microalgae Using Microbubble Interfacial Technology

**DOI:** 10.3390/bioengineering9120739

**Published:** 2022-11-29

**Authors:** Fahed Javed, Muhammad Waqas Saif-ul-Allah, Faisal Ahmed, Naim Rashid, Arif Hussain, William B. Zimmerman, Fahad Rehman

**Affiliations:** 1Microfluidics Research Group, Department of Chemical Engineering, COMSATS University Islamabad, Lahore Campus, Lahore 54000, Pakistan; 2Process and Energy Systems Engineering Center-PRESTIGE, Department of Chemical Engineering, COMSATS University Islamabad, Lahore Campus, Lahore 54000, Pakistan; 3Division of Sustainable Development, College of Science and Engineering, Hamad Bin Khalifa University, Qatar Foundation, Doha 34110, Qatar; 4Department of Chemical and Biological Engineering, The University of Sheffield, Sheffield S10 2TN, UK

**Keywords:** biodiesel, kinetics, esterification, RSM, microbubble technology

## Abstract

As an alternative to fossil fuels, biodiesel can be a source of clean and environmentally friendly energy source. However, its commercial application is limited by expensive feedstock and the slow nature of the pretreatment step-acid catalysis. The conventional approach to carry out this reaction uses stirred tank reactors. Recently, the lab-scale experiments using microbubble mediated mass transfer technology have demonstrated its potential use at commercial scale. However, all the studies conducted so far have been at a lab scale~100 mL of feedstock. To analyze the feasibility of microbubble technology, a larger pilot scale study is required. In this context, a kinetic study of microbubble technology at an intermediate scale is conducted (3 L of oil). Owing to the target for industrial application of the process, a commercial feedstock (*Spirulina*), microalgae oil (MO) and a commercial catalyst para-toluene sulfonic acid (PTSA) are used. Experiments to characterize the kinetics space (response surface, RSM) required for up-scaling are designed to develop a robust model. The model is compared with that developed by the gated recurrent unit (GRU) method. The maximum biodiesel conversion of 99.45 ± 1.3% is achieved by using these conditions: the molar ratio of MO to MeOH of 1:23.73 ratio, time of 60 min, and a catalyst loading of 3.3 wt% MO with an MO volume of 3 L. Furthermore, predicted models of RSM and GRU show proper fits to the experimental result. It was found that GRU produced a more accurate and robust model with correlation coefficient R^2^ = 0.9999 and root-mean-squared error (RSME) = 0.0515 in comparison with RSM model with R^2^ = 0.9844 and RMSE = 3.0832, respectively. Although RSM and GRU are fully empirical representations, they can be used for reactor up-scaling horizontally with microbubbles if the liquid layer height is held constant while the microbubble injection replicates along the floor of the reactor vessel—maintaining the tessellation pattern of the smaller vessel. This scaling approach maintains the local mixing profile, which is the major uncontrolled variable in conventional stirred tank reactor up-scaling.

## 1. Introduction

The renewable energy generation is an important factor in reducing harmful effects on the environment caused by the excess use of conventional fuels. Biodiesel derived from inexpensive feedstocks, such as microalgae, can be a suitable alternative for replacing fossil fuel. Biodiesel is generally produced from various resources, such as waste cooking oil, animal fats, energy corps, yeast lipids, and microalgae [1,2]. Biodiesel consists of long chains of carboxylic acids of alkyl ester produced from esterification and transesterification of lipids or oils [3,4].

For sustainable production of biodiesel, feedstock cost is the key parameter affecting overall economics of the process. Since biodiesel production through refined oils feedstocks (soybean oil or sunflower oil) is expensive, low-cost unrefined oil (animal or microalgae oil) can be utilized as a cheaper substitute for biodiesel production [5,6,7]. However, the unrefined feedstock’s generally consists of a large quantity of FFA. The presence of these FFA in feedstock are not suitable for transesterification using base catalysts due to the soap formation [8,9]. Therefore, acid catalyzed esterification is employed to reduce FFA content by converting them into FAMEs [10,11]. The acid-catalysis is significantly slower than transesterification due to the low-miscibility of MeOH with oil reducing overall mass transfer; as a result reaction rate decreases. The slower reaction rate directly affects the overall economics of the process [12,13,14]. However, the development of an economical method for commercial biodiesel production would be a revolutionary milestone for the fuel industry. Large scale biodiesel production could address many global challenges, such as waste management, energy supply, and environmental pollution. The major challenges that hinder commercialization of biodiesel include (1) cost-intensive methods of acid catalyzed esterification of biodiesel feedstock, (2) inability of the present technologies to scale-up, (3) expensive two stage production process, i.e., acid catalysis followed by base catalysis, and (4) expensiveness of various feedstocks [15,16,17].

Recently, microbubble mediated mass transfer technology has proven to increase the reaction rate by injecting one of the reactants in the vapor phase [18,19,20]. Microbubbles provide a large interfacial area, low buoyancy force, and high contact time on the bubble surface, which facilitate the rate of reaction of the system [21,22]. Furthermore, a smaller radius causes the increase of pressure inside the microbubble, as stated by Young–Laplace’s law. Hence, the temperature inside the bubble could be predicted to be higher as compared with the boiling point of the alcohol. This also increases the surface energy of the bubble. All of these factors have yielded an unprecedented higher rate and conversion of the esterification reaction [18,20,23]. For example, Fahed et al. (2019) investigated the effect of microbubble on the esterification reaction by producing ethyl acetate and achieved 79.95% in 35 min compared with the conventional method, which achieved 64% conversion of esterification reaction in 350 min for ethyl acetate production [23]. In another study, Naveed et al. (2019) reported a 97% conversion of oleic acid into biodiesel in 30 min using microbubble technology which is a higher conversion than the conventional method, which achieved 80% conversion in 312 min using H_2_SO_4_ as a catalyst [20]. The major focus of these studies was to develop microbubble technology for esterification reaction using single component feedstock. In this context, Fahed et al. (2021) validated the effectiveness of microbubble technology in an unrefined feedstock (chicken fat oil) and showed an overall process conversion of 89.90% in 30 min [18]. To further investigate the effect of microbubble technology, Fahed et al. (2022) investigated the effect of microbubble technology by integrating microbubble technology with heterogeneous catalyst using waste cooking oil to further increase the rate of reaction and achieved an 85% conversion in 20 min [24]. The higher conversion was achieved for both chicken fat oil and waste cooking oil in a shorter period of time, indicating that microbubble technology is an economical method for biodiesel production.

However, all previous studies on microbubble technology mainly focused on lab-scale experiments~100 mL of oil and without significant control over vapor pressure of MeOH. The major focus of this study is to scale-up the microbubble mediated mass transfer technology from lab-scale to semi-pilot scale with up to 3 L volume of oil. The experiments were design using response surface methodology (RSM) and compared with gated recurrent unit (GRU). There are many studies in the literature that use RSM for optimization for biodiesel production [25,26]. However, the current study is the first study that implemented both RSM and GRU using microbubble technology. Furthermore, several technologies have been developed to manipulate the reaction equilibrium to achieve a higher conversion. These reactions are usually slow and limited by reaction kinetics and mass transfer, which are key constraints, such as esterification reactions. However, an entirely different technique has been developed using microbubble mediated mass transfer technology: an inherent liquid–liquid reaction is converted into liquid–vapor reaction, which entirely changes the reaction kinetics. The kinetics of a reaction are entirely based on two hypotheses, i.e., (1) by increasing interfacial area, mass transfer of the process increases to which rate of reaction is also enhanced. (2) Simultaneous removal of a reactant could also increase the reaction kinetics in the forward direction. Moreover, this study also investigated reaction kinetics on the semi-pilot scale the first time to understand the feasibility and compatibility of microbubble technology for further scale-up.

Keeping the semi-pilot scale nature of the study, a commercial feedstock (biomass of *Spirulina*) and a commercial catalyst p-Toluenesulfonic acid (p-TSA) is chosen to investigate microbubble mediated mass transfer. Oil from *Spirulina* was derived using solvent extraction method. Experiments were designed using response surface methodology (RSM), a robust model derived from RSM compared with another model developed using gated recurrent unit (GRU). This is the first study that demonstrated the potential scale-up of microbubble technology and compared both RSM and GRU models to increase the commercial feasibility of the process.

## 2. Material and Methods

### 2.1. Materials

*Spirulina*-biomass was purchased from Sentron Asia Company in Lahore, Pakistan. p-Toluenesulfonic acid (p-TSA) was purchased from Sigma Aldrich, St. Louis, MO, USA. MeOH of 99% analytical grade and 99% analytical grade n-Hexane were purchased from DAEJUNG chemicals, Siheung-si, South Korea.

### 2.2. Lipids Extraction from Spirulina Biomass

Microalgae oil/lipids (MO) were extracted from dry biomass using hexane and MeOH in a ratio of 7:3 by vol %. The biomass of *Spirulina* and solvent were stirred at 1000 rpm for 6 h at room temperature. Afterward, biomass was separated through filtration (whatman filter paper 42), and oil was recovered by evaporating solvent using vacuum evaporation (Buchi R-210, BUCHI Corporation, New Castle, DE, USA) at 60 °C [27]. The gravimetric method determined the oil yield [27]. Physiochemical properties and lipid composition of derived MO are presented in Table 1.

### 2.3. Pilot-Scale Experimental Setup for Biodiesel Production

The esterification reaction between MO and MeOH was performed using p-TSA as a catalyst. The schematic diagram of the pilot scale process was shown in Figure 1. In the current pilot-scale process, MeOH vapors were formed using a local fabricated digitally controlled vaporizer purchased from EES Technologies, Lahore, Pakistan. The vaporizer provided MeOH vapor at control/desired vapor flowrate, pressure, and temperature. A customized bubble reactor was fabricated using grade 3 sintered borosilicate glass diffuser. The total volume of the bubble reactor was 3.5 L (radius = 43.18 mm and height = 609 mm), and the working volume was up to 3 L. The experiments were designed using RSM (Design Expert 11). The details of the RSM model are given in the next section. The bubble reactor was filled with different volumes of microalgae MO according to the response surface methodology RSM model. The MeOH vapors were produced in a vaporizer then passed through a borosilicate glass diffuser to form microbubbles. The temperature of the reactor was measured through a thermocouple (Digital thermometer, Jiangsu, China). The sample was collected continuously at a regular interval of 10 min. Once, the reaction had been run for the given time, the samples were filtered (whatman filter paper 42) and washed with deionized water. The samples were dried using a vacuum evaporator (Buchi R-210, BUCHI Corporation, New Castle, DE, USA) and stored for further analysis. All the experiments were performed in triplicate, and their average values with standard deviation was reported.

### 2.4. Modeling and Experimental Design through Response Surface Methodology

RSM is a statistical and mathematical tool that uses multiple variables to design experiments for optimization [28]. In the current study, BBD was used to design experiments. BBD was used to design a process with more than two factors; this method provides fewer experiments than factorial design. Furthermore, BBD follows a cubical design edge using a midpoint with three levels each (−1, 0, +1) [29].

BBD was used with three factors and five center points in the current study. Three factors used in this study are: *A* (molar ratio of oil to MeOH = 1:5 to 1:25), *B* (catalyst dosage = 0 to 5 g wt% of MO), and *C* (TIme-10 to 90 min). According to this design, a total of 17 runs were conducted to evaluate the current process feasibility. The designed experiments and their response with predicted values of RSM and GRU are shown in Appendix A and the RSM final conversion equation with coded value was given in Equation (1) [30,31]. The goodness-of-fit summary provided by RSM shows that quadric model is best suited for current experimental design Appendix A. The suggested model is best suited for current experimental responses, and the current suggested model is also assessed through analysis of variance (ANOVA) as shown in Appendix A.
(1)$$Conversion=88.68+3.27A+26.13B+7.97C+3.24AB+3.49AC+3.98BC-1.74A^{2}-29.76B^{2}-5.81C^{2}$$

### 2.5. Modeling through Gated Recurrent Unit (GRU)

Commonly used artificial neural networks (ANN) involve three main layers, such as (1) input layer, (2) hidden layer (3) output layer and are indicated as *x*, *h*, and *y*. Recurrent neural network, a variant of ANN, was proposed for modeling time series and sequential data [32]. However, the vanishing gradient issue with large sequential data limits the application of RNNs. To solve the vanishing gradient problem, long short-term memory (LSTM), a variant of RNN, was introduced by Hochreiter and Schmidhuber [33]. This new variant of RNN, LSTM, was incorporated with four gates to replace the original hidden state in the memory cell. In 2014, GRU was introduced by adding a gated mechanism to the recurrent neural network [34]. Unlike LSTM, GRU merged the input gate and the forget gate into the update gate, as shown in Figure 2.

Hidden unit activation ($r_{t}$) is processed at a time step using Equation (2):(2)$$r_{t}=\sigma \left(W_{r}h_{t-1}+U_{r}x_{t}\right)$$

Here, σ depicts logistic sigmoid function, $W_{r}$ and $U_{r}$ represents weight matrices. After that, by using tanh type layer $\tilde{h_{t}}$ is calculated using $r_{t}$:(3)$$\tilde{h_{t}}=\tan h\left(W\left(r_{t}\times h_{t-1}\right)+Ux_{t}\right)$$

Equation (4) is the equation that distinguishes GRU from the LSTM. Here, $z_{t}$ combined the remember gate along with forget gate in LSTM. $z_{t}$ is calculated as follows:(4)$$z_{t}=\sigma \left(W_{z}h_{t-1}+U_{z}x_{t}\right)$$

Lastly, the hidden state ($h_{t}$) in GRU is calculated using Equation (5):(5)$$h_{t}=\left(1-z_{t}\right)\left(h_{t-1}\right)+\left(z_{t}\right)\left(\tilde{h_{t}}\right)$$

This study incorporated artificial intelligence modeling for the prediction of response surface, conversion in this case. Gated recurrent unit (GRU) was utilized as a sequence learning deep learning technique that requires suitable value of its hyper-parameters. Suitable architecture of the GRU model was obtained by varying hyper-parameters values (Table 2). To study the comparison between GRU and RSM, different parameters were considered, such as R^2^, RMSE, MAPE, and mean absolute error MAE.

### 2.6. Biodiesel Analysis

Biodiesel analysis was performed using gas chromatography (GC) and ASTM method of FFA analysis. Briefly, in GC (Shimadzu GC-2014, Shimadzu Europa, Duisburg, Germany) the system was equipped with a flame ionization detector with column EN14103 (30 m × 0.32 mm id. × 0.25 µm film thickness). Nitrogen was introduced as a carrier with an initial temperature of 523 K and a split ratio of 50:1 [20]. For FFA analysis, the AOCS standard titration method was used [36,37]. To determine the FFA of the solution, the following Equations (6) and (7) were used [27]:(6)$$\mathrm{Acid}\;\mathrm{value}\left(\frac{\mathrm{mgKOH}}{g\;\mathrm{biodiesel}}\right)=\frac{\left(F_{A}-F_{B}\right)\times \;N\;\times 56.11}{W}$$
(7)$$\;\mathrm{Free}\;\mathrm{fatty}\;\mathrm{acid}\;\left(\mathrm{FFA}\right)\;\left(\%\right)=\frac{1}{2}\times \mathrm{AV}$$

## 3. Results and Discussion

### 3.1. Effect of Different Parameters on Free Fatty Acid Conversion

#### 3.1.1. Effect of Catalyst Loading and Molar Ratio on Free Fatty Acid Conversion

The simultaneous effect of the molar ratio of oil: MeOH and catalyst loading are shown in Figure 3. The graph indicates that increasing catalyst loading from 0 to 5 wt% of MO significantly increases the conversion of the process. Increasing the catalyst loading enhances the protonation of FFA in MO. As the degree of protonation increases, the conversion of FFA and the reaction rate also increases. A further increase in catalyst loading after a certain point conversion of FFA was not increased due to insufficient active sites of MO. Furthermore, Perturbation plot of RSM indicate that most dominate factor in the current study is catalyst loading as show in Appendix A.

On the other hand, by increasing the molar ratio of oil and MeOH, the FFA conversion also increases due to the contact time of FFA with MeOH vapors increasing. At a lower molar ratio, less volume of MeOH passes through the MO and leaves the system and vice versa. However, increasing the molar ratio after a certain limit, a small amount of MeOH can start to accumulate in the reactor, slightly reducing the reaction rate, as indicated by the results. RSM’s optimized condition was molar ratio: 1:23.73 Oil: (MeOH) and catalyst loading 3.3 wt% of MO.

#### 3.1.2. Effect of Reaction Time and Molar Ratio on Free Fatty Acid Conversion

The effect of both time and molar ratio was investigated and are shown in Figure 4. Time is another important parameter to study the rate of reaction. The results indicate that increasing time conversion and the rate of reaction increase due to an increase in reaction time increases the contact time of oil molecules with MeOH. As a result, conversion of FFA increases. However, a higher molar ratio and less reaction time decrease the reaction rate due to the bubble having less contact time with MeOH, as the flowrate of MeOH vapors is too high. As a result, the result MeOH vapors leave the system unreacted [18,20,23].

Furthermore, an increase in MeOH flowrate also increases the formation of macrobubbles at higher flow rates and tends to produce larger bubbles [38]. Macrobubbles have a large buoyancy-force and less residence time compared to microbubbles. As they rise significantly faster, MeOH does not come into contact with FFA and, as a result, MeOH leaves the system unreacted. Both optimize reaction time and molar ratio can also provide an optimized flow rate to increase FFA conversion. The optimized time and molar ratio according to RSM were molar ratios: 1:23.73 Oil: (MeOH) and Time = 59.79~60 min.

#### 3.1.3. Effect of Reaction Time and Catalyst Loading on Free Fatty Acid Conversion

The simultaneous effect of catalyst loading and reaction time were important parameters for scaling up of biodiesel process. These parameters greatly affect the cost and energy of the biodiesel process. The interactive effect of catalyst loading and reaction time on response surface are shown in Figure 5. FFA conversion tends to increase with the reaction time. However, the effect of reaction time is masked by the effect of catalyst loading. The 3D plot also shows that catalyst loading directly relates to FFA conversion due to an increase in catalyst loading increasing the number of available reaction sides; as a result, higher conversion of FFA was achieved.

### 3.2. Scale-Up of Microbubble Reactor

The main goal of this work is to investigate the scale-up capacity of microbubble technology at optimized conditions provided by RSM and GRU and verify that provided conditions are suitable for converting FFA into biodiesel. For scale-up of microbubble reactor, different experiments were performed at different volumes of reactor varying from 1 to 3 L (Conversion: 1 L = 99.12%, 2 L = 99.55%, 3 L = 99.45%), as shown in Figure 6. It was observed that an increase in the volume in the reactor has negligible effect on the FFA conversion. An increase in the volume of oil also increased the pressure head of the microbubble reactor. By increasing the volume of oil, microbubbles stay in contact with the oil for a longer period of time. However, it is observed that an equilibrium is achieved at 200 mm and does not change if it is increased any further. This could be explained on the basis of mass transfer occurring across the bubble interface and kinetics of the reaction. At 200 mm, a boundary layer would have formed at the bubble interface stopping the mass transfer of MeOH from inside the bubble to the oil working as a “solid sphere”. Several studies in hot microbubble distillation/stripping show that increasing the liquid layer heat can decrease separation efficiency because the non-equilibrium driving force diminishes [39]. Thus, further upscaling of the reactor should be horizontal by maintaining the microbubble tessellation pattern in the reactor. However, this scaling approach maintains the local mixing profile, which is the major uncontrolled variable in conventional stirred tank reactor upscaling.

Conventionally, biodiesel is produced by mixing both reactants in liquid phase in a batch reactor. The mass transfer is limited by low miscibility of reactants, apart from castor oil/MeOH, which has an OH at C-12, reducing the overall conversion. However, the current process MeOH was injected in the form of vapors (bubbles) and as a result a vapor–liquid system is formed. In vapor–liquid system, conversion of MO is also increased as diffusion of vapor–liquid is higher than the liquid–liquid reaction. Furthermore, microbubbles exhibit less buoyancy force, increasing the residence time of MeOH bubble. FFA is premixed with the catalyst and is already protonated. As the microbubbles of MeOH rise, the reaction between MeOH and protonated MO starts instantaneously. The amount of alcohol available in the interface is in excess as compared to the available MO pushing the reaction in a forward direction. As the bubble rises, the alcohol is transferred into the MO. As the temperature of the reactor is maintained higher than the boiling point of MeOH, it does not condenses in the reactor and leaves as vapor [24]. The vapors of MeOH can be collected, condensed and recycled to improve the process economics. Figure 7 clearly exhibits three different slopes. The highest rate is achieved in the first 10 min of the process. Afterword, the rate slightly slows down and an overall conversion of 97% is achieved in the next 30 min (overall 40 min). Since almost all of the MO has already reacted, only 2% conversion is achieved in the last 20 min. To enhance feasibility and hence the economics, the reaction could be stopped at 10 min and product could be separated using a suitable separation technique, such as distillation or else 99% pure product could be obtained in 60 min. Comparison of current MO study with other conventional acid catalyst based biodiesel production is shown in Table 3.

### 3.3. Reaction Kinetics of Biodiesel Conversion

Scale-up results already show that by increasing the liquid layer height, the conversion does not change and has been identified; the reaction kinetics for a microbubble mediated esterification system can be upscaled via scaling out the reactor horizontally, maintaining the same aerator pattern along the reactor bottom surface. This is a logical conclusion from the seminal paper of Al-Mashhadani et al. (2015) [41]. In their paper, the geometry of the placement of the internal baffle in an airlift loop reactor is systematically varied, yet the hydrodynamics of the phase distribution is invariant to the baffles position, basically demonstrating that it is insignificant. Only the height above the aerator injection point is shown to matter. This follows logically from the fact that microbubbles that are injected in laminar flow and maintain laminar flow only rise vertically. By tessellating the aerators to provide downcomer regions in between, this configuration replicates the micromixing and bulk mixing profiles. This horizontal tessellation/upscaling approach has been found in all pilot scale studies for microbubble distillation/hot microbubble stripping [39]. Rees-Zimmerman and Chaffin (2021) [42] found that in modeling the hydrodynamics of tall bioreactors with variable bubble size, there is no mass transfer limitation with microbubbles, but the reactor height above a fixed level is immaterial. Hence, studying the kinetics for this critical layer height is sufficient for upscaling horizontally.

The kinetics of this study were investigated under optimized condition of RSM, i.e., the molar ratio of MO to MeOH of 1:23.73 ratio, time of 60 min, and a catalyst loading of 3.3 wt% MO. The results show that the difference between predicted value of RSM (99.10% as shown in Appendix A) and actual value (99.45%) was less than 0.5% which indicate the validation of the current RSM model. To evaluate the reaction behavior of vapor–liquid, *Ha* was calculated using Equation (8) to assess the whether a reaction occurs on the bubble’s bulk or surface [43,44].
(8)$$Ha=\frac{\sqrt{{\left(M_{g/l}\right)}_{T}\;k\;C_{b}}}{k_{bl}}$$
(9)$${\left(M_{g/l}\right)}_{25℃}=6.02\times 10^{-5}\left(\frac{V_{l}^{0.36}}{\mu_{l}^{0.61}V_{g}^{0.64}}\right)$$
(10)$${\left(M_{g/l}\right)}_{T}=4.996\times 10^{3}{\left(M_{g/l}\right)}_{25℃}\exp\left(\frac{-2539}{T}\right)$$

*M_g/l_* was calculated at 25 and 70 °C by Equations (9) and (10) [45]. For a bubble size less than 2 mm, Equation (11) was used to calculate *k_bl_* [46].
(11)kbl=0.31((Dg/l)2ρlgμl)13

The calculated value of *Ha* is greater than 1, which indicates that the reaction occurs on the bubble surface, due to which bubble size is a crucial parameter of controlling reaction kinetics. The order of reaction was calculated using *E* by Equation (12) [43].
(12)$$E=Ha\left(1-\frac{Ha-1}{2E_{i}}\right)\;$$

*E_i_* was determined by using Equation (13) [43]
(13)$$E_{i}=1+{\left(M_{g/l}\right)}_{T}\left(\frac{C_{b}H}{b\;P_{g}}\right)\;$$

The value of *Ha* and *E* show that the current reaction is pseudo first order due to the value of both of them I almost equal. The rate of reaction was calculated by using Equation (14).
(14)$$\;\;-r_{A}=\frac{1}{\frac{1}{k_{g}\sigma }+\frac{H}{a\sqrt{{\left(D_{g/l}\right)}_{T}kC_{b}}}}P_{g}\;$$

The values used to calculate for *k_g_* σ, *k_bl_*, and *H* are 5.32 × 10^−3^ kmol s^−1^ m^−3^, 1.24 × 10^−4^ ms^−1^ and 43.05 kmol s^−1^.m^−3^ Pa, respectively. The final reaction rate was determine using Equation (15);
(15)$$-r_{A}=\left(1.32\times 10^{-5}\right)\left(P_{A}\times 101,325\right)\left(\sqrt{C_{b}}\right)$$

To determine the current activation energy (*E_A_*) in the scale-up reactor, the Arrhenius equation was used. The experiments were conducted by varying the reaction temperature (70–90 °C) [43]. The Arrhenius equation was used to develop the relations with rate constants to develop the equation to determine *E_A_* using Equation (16) [47]. An Arrhenius plot between *ln* (*k*) and 1/T is shown in Figure 8.
(16)$$\;ln\;k=-\frac{E_{A}}{RT}+ln\;A^{{}^{\circ}}$$

The E_A_ of the esterification reaction in the current scale-up microbubble reactor is calculated to be 10.01 ± 0.3 kJ mol^−1^. The current E_A_ was significantly less than conventional processes, as shown in Table 4. This low E_A_ indicates that reactions occur on the surface of bubbles [20]. It also shows that less energy is needed for MO to pass the barrier to form a product. In addition, the latent heat of MeOH was also freely available as free energy, making the reaction nature more exergonic, enhancing the rate of reaction and reducing the E_A_. The current reaction kinetics and reduced E_A_ successfully show the implementation of current scale-up reactor to industrial level.

### 3.4. Gated Recurrent Unit and Response Surface Methodology Comparison

RSM model has reported 0.9844 correlation coefficient R^2^ that shows good fitting efficiency, as shown in Figure 9. However, GRU has shown superior efficiency (Figure 10) and reported 0.9999 R^2^. The values of mean absolute percentage error (MAPE), root mean square error (RMSE) and mean absolute error (MAE) for RSM model and GRU model have been reported in Table 5.

Table 5 confirms the superiority of the GRU model over RSM as its predicted values are in good agreement with actual values, as shown in Figure 10. It can also be confirmed that GRU reported performance criterions used, such as MAPE, RMSE, and MAE, lesser than that of the RSM model. From Figure 9, deviating RSM model prediction values from actual values are clearly visible, causing lower R^2^ (0.9844) compared to that of GRU (0.9999), as well as a reported larger MAPE, RMSE and MAE compared to that of the GRU model, as show in Figure 10. The MAE for the RSM model is 2.6847, which is approximately 60 times higher compared to that of the GRU model (0.045). Furthermore, MAPE for the GRU model (0.00083) confirms its superiority over the RSM model (0.0465). Furthermore, in terms of RMSE, GRU reported an RMSE of 0.0515 and RSM reported 3.0832.

Furthermore, a 45-degree perfect line was added to the plot to better understand the prediction accuracy of the models. The prediction performance increases as the scatter points approach toward a 45-degree perfect line and hence decreases the error. The plot depicted that the GRU model prediction of conversion% were following the 45-degree perfect line more strictly than that of the RSM model and hence reported lesser RMSE, MAPE, and MAE. Furthermore, a higher R^2^ value for the GRU model (0.9999) depicts outperforming performance compared to the R^2^ value for the RSM model (0.9844). This scatter plot comparison also pointed out that the GRU model has performed much better than the RSM model to predict conversion.

## 4. Conclusions

The current study successfully developed a robust model which shows a high feasibility of microalgae-based biodiesel on a semi-pilot scale. Both models were in line with experimental observation. In addition, the comparison of both RSM and GRU showed that GRU was more accurate than RSM to predict the conversion. Furthermore, predicted models of RSM and GRU show a proper fit to the experimental result. It was found that GRU produced a more accurate and robust model with correlation coefficient R^2^ = 0.9999 and root-mean-squared error (RSME) = 0.0515 in comparison with the RSM model with R^2^ = 0.9844 and RMSE = 3.0832, respectively. Furthermore, the kinetics of the pilot scale microbubble reactor revealed that more than 99.45 ± 1.3% conversion of FFA into biodiesel was achieved in 60 min, and the current reaction follows pseudo first order kinetics with respect to MO. Additionally, a lower *E_A_* of 10.01 ± 0.3 kJ mol^−1^ indicates that less energy was required for reactant to jump the barrier to form product. Although RSM and GRU are fully empirical representations, they can be used for reactor upscaling horizontally with microbubbles. Horizontal out-scaling maintains the liquid layer height, while the microbubble injection replicates along the floor of the reactor vessel—keeping the tessellation pattern of the smaller vessel with the same bubble flux per unit area. This scaling approach maintains the local mixing profile, which is the major uncontrolled variable in conventional stirred tank reactor upscaling. This study should prove to be a milestone in future studies for further scale-up of microbubble mass transfer technology. Further research needs to be carried out in terms of reactor development and finding new material for reactor formation, keeping the process cost to a minimum. Additionally, life cycle analysis should be carried out to calculate the environmental impact of the microbubble technology. Life cycle analysis will provide overall insight toward sustainability of the developed approach to analyze manufacturing, ecological effect, and energy expenditures of current technology.

## Figures and Tables

**Figure 1 bioengineering-09-00739-f001:**
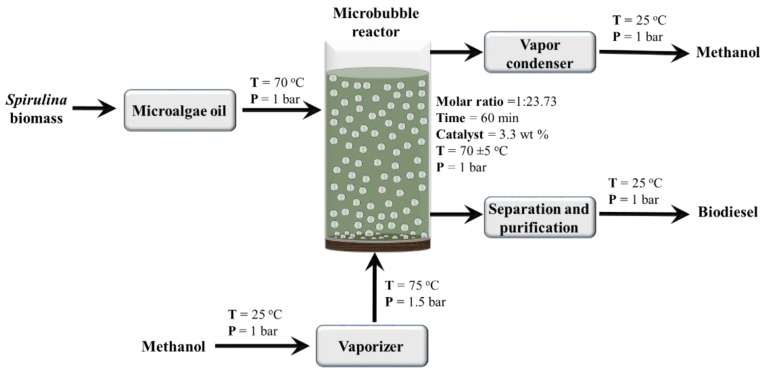
Process flow diagram of scale-up microbubble reactor.

**Figure 2 bioengineering-09-00739-f002:**
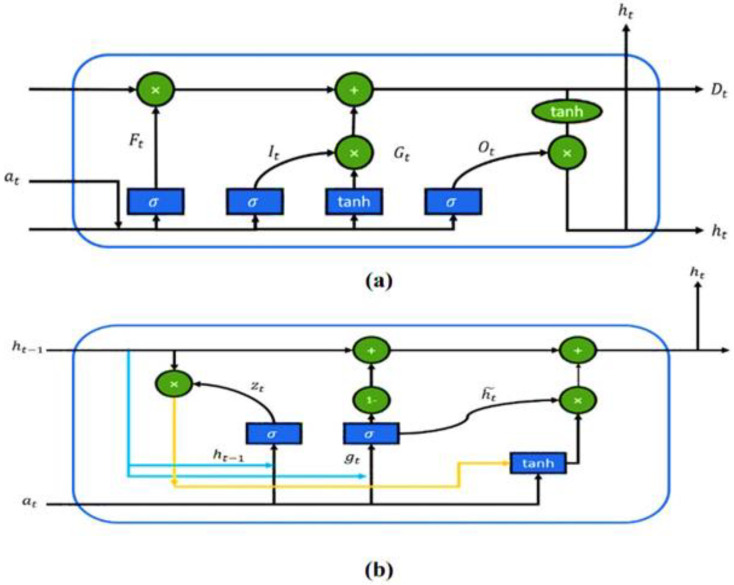
Comparison parameter between (**a**) LSTM and (**b**) GRU framework [35].

**Figure 3 bioengineering-09-00739-f003:**
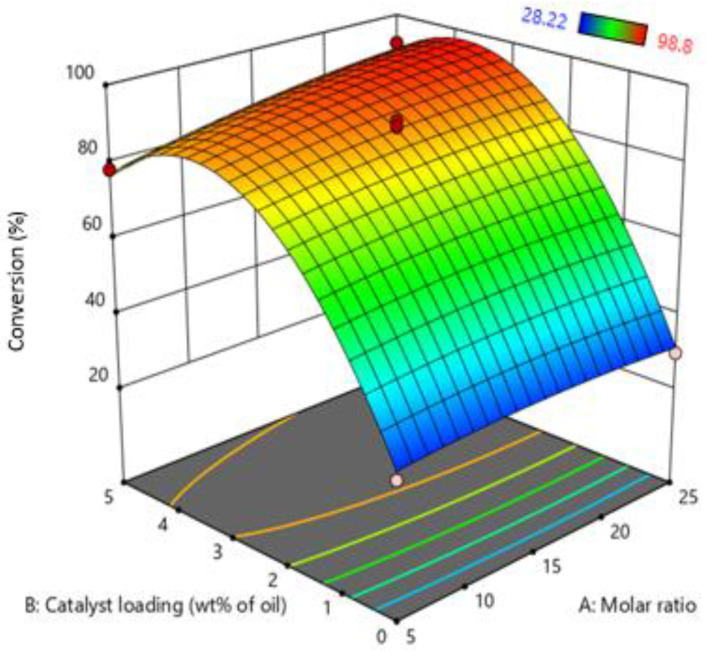
Effect of molar ratio (oil: MeOH) and catalyst loading on biodiesel production.

**Figure 4 bioengineering-09-00739-f004:**
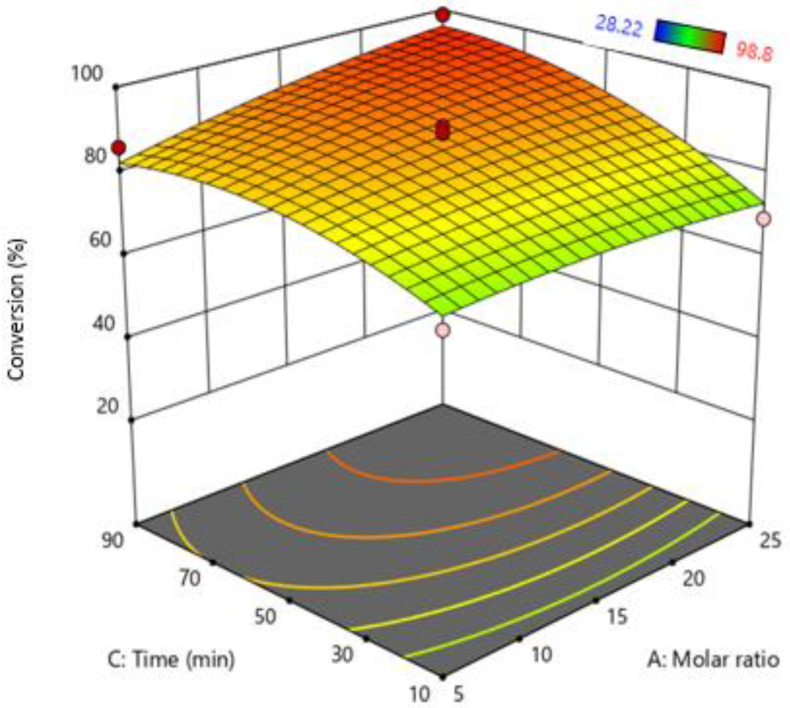
Interactive effect of molar ratio and reaction time.

**Figure 5 bioengineering-09-00739-f005:**
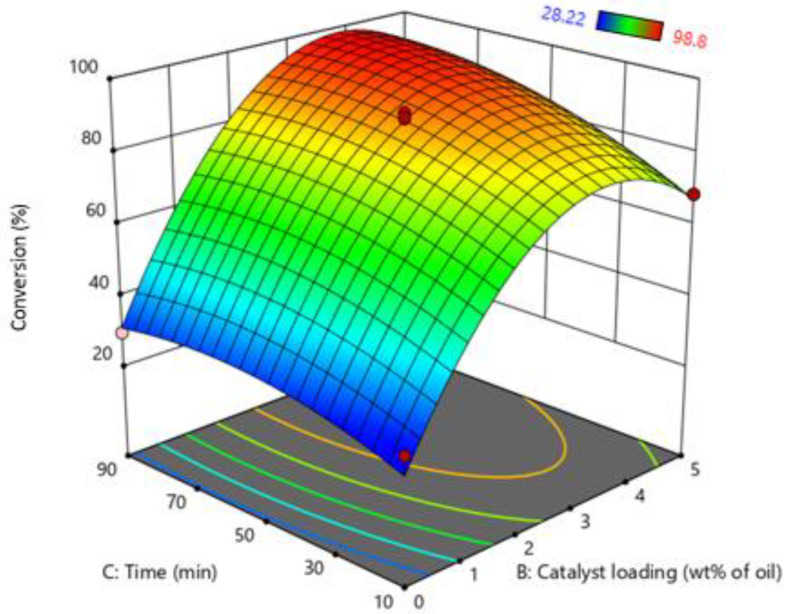
Interactive effect of reaction time and catalyst loading.

**Figure 6 bioengineering-09-00739-f006:**
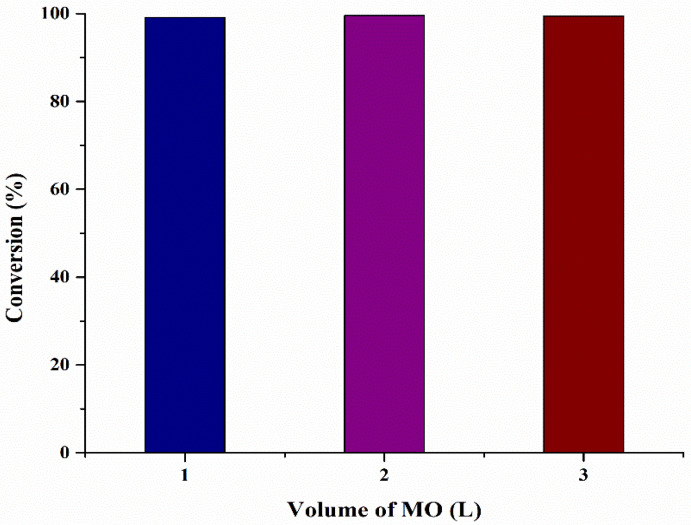
Effect on reaction conversion by increasing microalgae oil volume.

**Figure 7 bioengineering-09-00739-f007:**
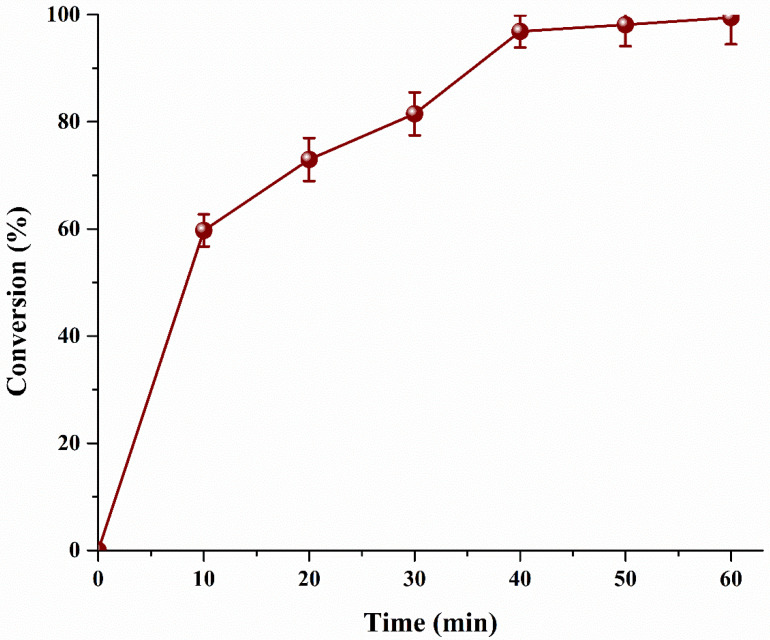
Effect microalgae oil conversion with respect to reaction time.

**Figure 8 bioengineering-09-00739-f008:**
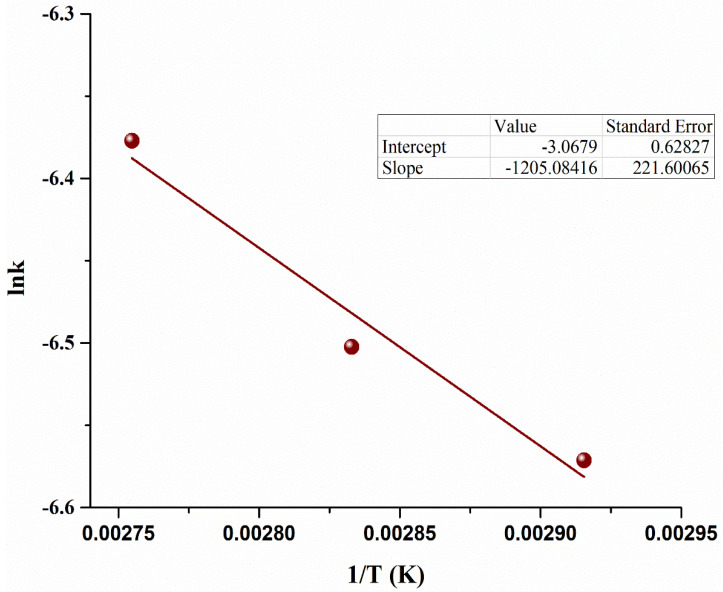
Arrhenius plot for microalgae oil based biodiesel reaction.

**Figure 9 bioengineering-09-00739-f009:**
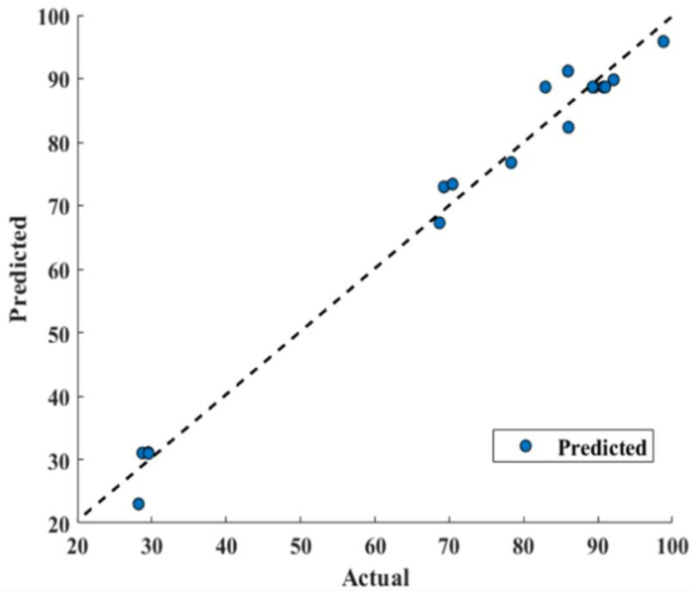
Response surface methodology model performance prediction (actual versus predicted).

**Figure 10 bioengineering-09-00739-f010:**
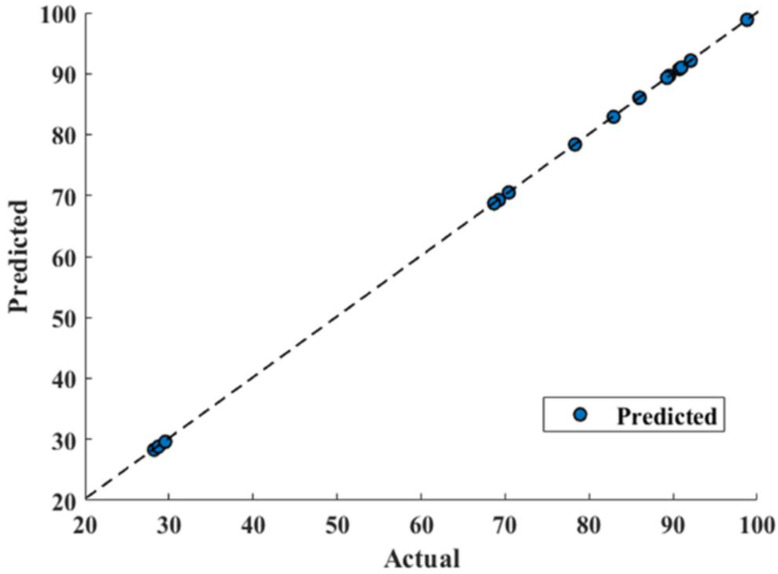
Gated recurrent unit model prediction.

**Table 1 bioengineering-09-00739-t001:** Physicochemical properties of microalgae oil.

Parameters	Units	Value
FFA content	%	32.5 ± 2
Density (25 °C)	Kg m^3^	920 ± 5
Kinematic viscosity (40 °C)	mm^2^ s^−1^	30.06 ± 3
**MO composition**		
Mystic acid (C14:0)	%	1.90 ± 0.5
Palmitic acid (C16:0)	%	35.67 ± 3
Palmitoleic acid (C16:1)	%	6.11 ± 2
Linoleic acid (C18:2)	%	48.55 ± 2
Linolenic acid (C18:3)	%	2.17 ± 0.5
Stearic acid (C18:0)	%	5.60 ± 2

**Table 2 bioengineering-09-00739-t002:** Hyperparameters for gated recurrent unit model.

Hyperparameters	Bounds	Set Values
Number of hidden units	Positive integers	50
Gradient threshold	0–1	0.1
Initial learning rate	0–1	0.01
Learn rate drop factor	0–1	0.2
Learn rate drop period	Positive integers	100
Training Epochs	Positive integers	150

**Table 3 bioengineering-09-00739-t003:** The comparison of current up-scaling study with other conventional catalyst based biodiesel production.

Feedstock	Catalyst	Reaction Time (min)	Conversion (%)	Reference
Conventional method	H_2_SO_4_	120	78	[40]
Microbubble Technology	H_2_SO_4_	30	98	[20]
Microbubble Technology	p-TSA	30	97	[20]
Microbubble Technology	p-TSA	30	89.90	[18]
Microbubble Technology	Sr/ZrO_2_	20	85	[24]
Microbubble Technology (semi pilot-scale)	p-TSA	60	99.45 ± 1.3	This study

**Table 4 bioengineering-09-00739-t004:** Comparison of activation energy with different biodiesel feedstocks.

Feedstock	Method	Scale of Experiments	Catalyst	E_A_ (kJ mol^−1^)	Reference
Jatropha	Conventional method	Lab-scale	1% H_2_SO_4_ and 1% NaOH	87.808	[48]
Microalgae	Supercritical method	Lab-scale	No catalyst	105	[49]
*Chlorella*	Conventional method	Lab-scale	HCl	38.892	[50]
*Spirulina platensis*	Single stage extraction–transesterification process	Lab-scale	H_2_SO_4_	14.518	[51]
Oleic acid	Microbubble technology	Lab-scale	7% H_2_SO_4_	26.37	[20]
Chicken fat oil	Microbubble technology	Lab-scale	7% PTSA	24.9	[18]
Spirulina	Microbubble technology	Semi pilot scale	3.3% PTSA	10.01 ± 0.3	This study

**Table 5 bioengineering-09-00739-t005:** Performance criteria comparison (response surface methodology vs. gated recurrent unit).

Criteria	Conversion % Prediction Performance	
	RSM Model	GRU Model
R^2^	0.9844	0.9999
MAPE	0.0465	0.00083
RMSE	3.0832	0.0515
MAE	2.6847	0.045

## Data Availability

The data presented in this study are available on request from the corresponding author. The data are not publicly available due to confidentiality of work.

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
