# Peer review of "Kinetics of Biodiesel Production from Microalgae Using Microbubble Interfacial Technology"

_bioengineering, 2022, doi:10.3390/bioengineering9120739_

Round 1
Reviewer 1 Report
This manuscript shows biodiesel production from algae using microbubble technology. The research is interesting however following issues must be addressed.
Why this study is novel? Microbubble technology? The authors should justify by citing related studies.
How the produced biodiesel is comparable with other used technology? why you obtained a 99.45% of conversion rate?
Is this the first study using this technology and RSM and GRU at the same time? please justify with previously published studies in this field.
Figure 1 needs more detail.
Is there any by-product such as glycerol?
The cited studies in Table 3 are too old and not acceptable for comparison. That's why I said you must compare your results with recently published studies only.
Figure 8 is not necessary.
The conclusion is not enough. The most significant results must be presented here.
Reviewer 2 Report
This paper can be accepted after the following minor revisions:
- Please include further results and conclusions in the abstract, right now it is very poor.
- Please include more background in the Introduction section so the readers can understand better the novelty of your paper.
- Please include further future works in the conclusion section
- Please upgrade English grammar. This point is mandatory.
Reviewer 3 Report
Always use at least triplicates data for experimental work. Thanks

Round 2
Reviewer 1 Report
The paper is modified accordingly and can be accepted.
Reviewer 3 Report
I have passed through the updated version of research manuscript and cover letter. Yes, the minors comments have been addressed, but I still have doubts about oil extractions, oil analysis methods and triplicates data.